# Dietary Patterns and Fibre Intake Are Associated with Disease Activity in Australian Adults with Inflammatory Bowel Disease: An Exploratory Dietary Pattern Analysis [note 1]

**DOI:** 10.3390/nu16244349

**Published:** 2024-12-17

**Authors:** Denelle Cosier, Kelly Lambert, Karen Charlton, Marijka Batterham, Robert D. Little, Nan Wu, Paris Tavakoli, Simon Ghaly, Joseph L. Pipicella, Susan Connor, Steven Leach, Daniel A. Lemberg, Yashar Houshyar, Thisun Jayawardana, Sabrina Koentgen, Georgina L. Hold

**Affiliations:** 1School of Medical, Indigenous and Health Sciences, University of Wollongong, Wollongong, NSW 2500, Australia; 2Statistical Consulting Centre, National Institute for Applied Statistical Research Australia, University of Wollongong, Wollongong, NSW 2500, Australia; 3Department of Gastroenterology, Alfred Health, Melbourne, VIC 3004, Australia; 4Faculty of Medicine, Nursing and Health Sciences, Monash University, Clayton, VIC 3800, Australia; 5University of New South Wales Microbiome Research Centre, Faculty of Medicine and Health, University of New South Wales, Sydney, NSW 2033, Australia; 6Department of Gastroenterology, Sutherland Hospital, Sydney, NSW 2229, Australia; 7Department of Gastroenterology and Hepatology, St Vincent’s Hospital Sydney and St Vincent’s Clinical School, UNSW Medicine & Health, University of New South Wales, Sydney, NSW 2033, Australia; 8Department of Gastroenterology, Liverpool Hospital and South West Sydney Clinical Campuses, UNSW Medicine & Health, University of New South Wales, Sydney, NSW 2033, Australia; 9Crohn’s Colitis Cure, Sydney, NSW 2009, Australia; 10Discipline of Paediatrics, School of Clinical Medicine, University of New South Wales, Sydney, NSW 2033, Australia; 11Department of Gastroenterology, Sydney Children’s Hospital, Sydney, NSW 2031, Australia

**Keywords:** inflammatory bowel disease, dietary pattern, faecal calprotectin, Crohn’s disease, ulcerative colitis, diet, principal component analysis, cluster analysis

## Abstract

Background: Few studies have explored the relationship between habitual dietary patterns and disease activity in people with Inflammatory Bowel Disease (IBD). This cross-sectional study explored the association between dietary patterns and clinical and objective markers of inflammation in adults from the Australian IBD Microbiome Study. Methods: Dietary patterns were derived using principal component analysis (PCA) of baseline food frequency questionnaire data. Food intake was quantified using 3-day food record data. Associations between dietary intake and both clinical disease activity index (CDAI) and faecal calprotectin (FCP) were analysed. Results: Participants included 412 adults (IBD = 223, Healthy controls (HC) = 189). Both cohorts consumed poor-quality diets with inadequate servings of most food groups compared to Australian reference standards. IBD participants without FCP inflammation had significantly higher fibre intake than those with moderate FCP. In the Crohn’s Disease group, high adherence to ‘High plant diversity’ and ‘Meat eaters’ dietary patterns were associated with increased CDAI and FCP, respectively. In the combined IBD cohort, high adherence to a ‘Vegan-style’ dietary pattern was associated with increased FCP. Conclusions: There is a need for dietary modifications among Australian adults, both with and without IBD, to improve dietary fibre intake and adherence to dietary guidelines. Dietary patterns characterised by a high intake of plant foods or meat products were both positively associated with indicators of active IBD. It is possible that some participants with active IBD were modifying their diet to try to manage their disease and reduce symptoms, contributing to the association between healthier dietary patterns and active disease. Further clinical and longitudinal studies are needed to expand upon the findings. This study offers a unique contribution by utilising FCP as an objective marker of intestinal inflammation and applying dietary pattern analysis to investigate the relationship between diet and inflammatory markers.

## 1. Introduction

Inflammatory Bowel Diseases (IBDs) are chronic, relapsing, gastrointestinal disorders affecting approximately 4.9 million people worldwide, with incidence increasing [1]. IBD consists of Crohn’s Disease (CD) and Ulcerative Colitis (UC) and is characterised by chronic inflammation that often requires long-term immunosuppressive therapy [2,3]. Epidemiological studies implicate diet in the risk of developing IBD, with the European Society for Clinical Nutrition and Metabolism recommending a diet rich in fruit, vegetables, and *n*-3 fatty acids; and low in *n*-6 fatty acids, for a decreased risk of developing IBD [4]. However, conclusive evidence supporting diet in the management of active disease is predominantly restricted to Exclusive Enteral Nutrition (EEN) and the Crohn’s Disease Exclusion Diet (CDED) with or without Partial Enteral Nutrition (PEN) [5]. Both of these diets are recommended in clinical practice to induce remission in CD, but the role of diet in the induction of remission in UC and the long-term management of both CD and UC is unclear [5].

Dietary pattern analyses enable exploration of whole-of-diet intake and its relationship to disease outcomes, rather than examining individual nutrients or foods [6]. Only three studies have described and explored the relationship between dietary patterns and clinical disease activity measures in adults with IBD [7,8,9]. Significant associations between ‘healthy’ dietary patterns and favorable disease outcomes were seen in one study [7]. While endoscopic and transmural assessment remain the gold standard for assessing disease activity and response to therapy in IBD, disease activity can be evaluated using non-invasive markers of inflammation including faecal calprotectin (FCP) [10,11]. No studies to date have investigated the association between dietary patterns and FCP levels in people with IBD. Given the exploratory nature of dietary pattern analyses, further research into dietary patterns and their association with clinically relevant markers of disease activity is needed to identify patterns of intake which are associated with improved disease outcomes.

Given the emerging understanding of the relationship between gut microbiota and IBD disease activity [12], dietary pattern research which investigates the habitual intake of foods and nutrients known to exert benefits on gut microbiota is needed in cohorts with IBD. A higher dietary fibre intake is associated with a reduced risk of developing CD [13], and dietary fibre intake has been shown to increase populations of beneficial bacteria in the gut microbiota of both healthy human populations and those with inflammatory diseases [14,15]. A few studies have explored the association between dietary fibre intake and disease activity in cohorts with IBD. Amongst an Indian cohort of 97 adults with IBD, dietary fibre intake was not a predictor of disease relapse over the 12-month study period [16]. Similarly, a cross-sectional study amongst 112 Australian adults with IBD quantified sub-categories of dietary fibre intake and found that clinical disease or intestinal inflammation measured through FCP was not associated with dietary fibre intake [17]. Further similar studies will be useful to confirm these findings and expand upon the evidence. Polyphenols, found in fruits, vegetables, tea, and coffee, have demonstrated anti-inflammatory and microbiota-modulating properties in humans; however, their habitual dietary intake has not been examined in cohorts with IBD [18,19,20]. Fermented foods also exert beneficial effects on the gut microbiome, particularly related to increased production of short-chain fatty acids (SCFAs) [21]. However, the intake and effect of fermented foods in people living with IBD have not previously been explored [22].

We identified a need for further studies to examine the association between habitual dietary patterns and clinically relevant measures of disease activity in cohorts with IBD, and to explore the relationship between dietary fibre, polyphenols, and fermented foods on disease activity in IBD. Therefore, the aims of this study were three-fold: (1) To quantify the habitual intake of dietary fibre, polyphenols, and fermented foods in adults with IBD and healthy controls; (2) To identify and describe dietary patterns using both principal component analysis (PCA) and cluster analysis (CA) techniques; and (3) To compare differences in dietary intake and dietary patterns between healthy vs. IBD cohorts (UC vs. CD), and active IBD vs. IBD in remission (as defined by clinical disease activity, FCP, and a composite measure of both).

This article is a revised and expanded version of a conference abstract entitled “Positive relationship between plant diversity dietary patterns and disease activity in Australian adults with Inflammatory Bowel Disease”, which was presented at the Nutrition Society of New Zealand and the Nutrition Society of Australia Joint Annual Scientific Meeting in Auckland on 28 November 2023 [23].

## 2. Materials and Methods

### 2.1. Study Sample

Data from the Australian Inflammatory Bowel Disease (IBD) Microbiome (AIM) cohort study were used for this cross-sectional analysis [24]. The AIM study is a prospective, longitudinal cohort study following participants over a 24-month period, collecting clinical data, biological samples, and patient-reported outcomes. The AIM study recruited participants between the ages of 6–80 years who fall into one of five groups—Crohn’s Disease (CD), Ulcerative Colitis (UC), IBD-undifferentiated (IBD-U), healthy first-degree relatives of patients with IBD, and population healthy controls [24]. Participants in the healthy control (HC) groups were eligible if they did not have a history of irritable bowel syndrome, autoimmune disorders, or previous bowel surgery [24]. In this exploratory study, the sample consisted of adult participants who submitted dietary records between 1 July 2019 and 30 December 2022.

### 2.2. Dietary Data

AIM study participants completed a 3-day food record at baseline, collected at the same time as clinical and biomarker data. The food record was used to calculate the intake of food groups, dietary fibre, polyphenols, and fermented foods. Food record data were exported into FoodWorks 10 Professional Nutrient analysis software (Version 10; Xyris Australia Pty Ltd., Brisbane, Australia). for nutrient analysis. The multiple source method (MSM) was used to calculate the habitual dietary intake data from the 3-day food records [25]. Duplicate and incomplete (less than 3 days) dietary records were deleted. Dietary records containing >3 days of data were shortened to the first 3 days of recording. AIM participants also completed the Victorian Cancer Council 80-item Food Frequency Questionnaire (FFQ) [26] at baseline. This FFQ has been validated to accurately capture dietary intake in various Australian populations [27,28]. FFQ data were used to identify dietary patterns using principal component analysis (PCA) and cluster analysis (CA) methods. The rationale for using FFQ data for the dietary pattern analyses instead of 3-day food records was to enhance comparability to previous dietary pattern studies which used FFQ data [7,8,9]. Additionally, the output of the FFQ data is provided in a form that is much more suited to grouping for PCA than 3-day food record data. See Figure 1 for a graphical summary of how the dietary data were transformed and utilised in this study. Only participants with a complete 3-day food record and FFQ were included in this study. Under- and over-reporters of dietary intake were excluded using cutoffs of <500 kcal/day and >3500 kcal/day, respectively [29]. To perform PCA and CA on the FFQ data, the 80 food items were collapsed into 32 food groups aligning with the Australian Dietary Guidelines [30] in order to provide a manageable number of principal components for the analysis [8,31,32]. The categorisation of food items is shown in Appendix A.

### 2.3. Dietary Fibre, Polyphenol, and Fermented Food Identification

To obtain accurate food group intake data, the 3-day food record data were merged with the Australian Dietary Guideline classification system developed by Food Standards Australia New Zealand [33] (see Appendix A). To categorise fibre intake into insoluble fibre, soluble fibre, and resistant starch, the Australian Fibre Categories Database (FCD) by Fuller et al. (2018) [34] was merged with the 3-day food record data. Polyphenol content of foods was calculated using the Phenol-Explorer 3.0 database [35] and, when data were unavailable, the USDA Database for the Flavonoid Content of Selected Foods [36] was used. All major polyphenol classes were calculated, namely, flavonoids, phenolic acids, stilbenes, and lignans. Fermented foods and drinks were identified according to the ISAPP definition—as those that are “made through desired microbial growth and enzymatic conversions of food components” [22] (see Appendix A for the identified fermented foods and drinks). Each occasion of fermented food/drink consumption was assigned to one of the listed fermented food categories by manually examining food records. Probiotic supplements, indicated on participants’ demographic surveys, were also assigned to the probiotic column, and assigned a value. The sum of each fermented food category and the total fermented food intake was calculated for each participant. Detailed information on the steps undertaken to merge the food groups dataset and FCD to the 3-day food record data is provided in Appendix A.

### 2.4. Demographic and Clinical Data

The following demographic data were analysed: sex, date of birth, country of birth, highest qualification, anthropometry (height, weight, Body Mass Index (BMI)), smoking status, IBD subtype, age at diagnosis, disease duration, disease distribution and phenotype, presence of extra-intestinal manifestations (EIMs), surgical history, and current medications. Clinical disease activity was assessed using the partial Mayo score or Crohn’s Disease Activity Index (CDAI). Partial Mayo scores were categorised as 0–1 (clinical remission), and 2 (clinically active disease) [37]. A CDAI score of >150 is typically considered to indicate clinically active disease in clinical research, therefore CDAI was categorised as <150 (clinical remission) or >150 (clinically active disease) [38,39].

### 2.5. Biomarker Data

FCP data were analysed for all 211 samples collected. FCP was analysed using CalproLab ELISA kits and values ≤20 μg/g were replaced with the value of 0 μg/g according to the manufacturer’s standard reporting [40]. An FCP value <50 μg/g has previously been reported as a suitable cutoff to distinguish between intestinal inflammation indicating possible IBD and functional gastrointestinal disorders [41,42] and was used to indicate no intestinal inflammation in this cohort. Previous research has shown variation in FCP cutoffs specific for detecting disease remission and alternatively predicting disease activity and relapse, with cutoffs ranging from <100 to <250 μg/g [41,43,44,45]. To optimise the distribution of sample sizes in this study, we categorised FCP 50–150 μg/g as mild intestinal inflammation, and FCP > 150 μg/g as high intestinal inflammation. For a closer examination of IBD disease activity, a composite classification was considered using both FCP and clinical disease scores. This composite classification considered participants in remission with: (1) a partial Mayo < 2 or CDAI < 150 and (2) FCP ≤ 150 μg/g. Participants were classified as in active disease according to the composite classification if they had FCP data and had (1) a partial Mayo ≥ 2 or CDAI ≥ 150 or (2) FCP > 150 μg/g.

### 2.6. Determination of Dietary Patterns Using Principal Component Analysis

PCA was used to identify dietary patterns based on correlations between the 28 food groups from the FFQ data. PCA was performed separately for participants with IBD and healthy controls. PCA reduces the dimensionality of food intake variables to fewer variables without losing information [46]. Input variables are grouped according to how they correlate with each other, creating distinct patterns of food variables correlated together from the dietary data [47]. A Kaiser–Meyer–Olkin (KMO) test score of ≥0.5 for the complete model was accepted to confirm the assumption of sampling adequacy, and a significant Bartlett’s test of Sphericity (*p* ≤ 0.05) was accepted to confirm the assumption of data suitable for reduction. Variables with a KMO test score of ≥0.5 in the anti-image matrix were retained in the model [48]. The number of dietary patterns to retain was determined based on an eigenvalue of >1 and identification of a break in the scree plot [49]. The PCA was orthogonally rotated to increase interpretability. Food variables with a factor loading ≥|0.30| were included in the component matrix [50,51]. Factor loadings with positive values indicate the corresponding food variable is positively associated with the dietary pattern, and negative factor loadings indicate an inverse association. The larger the factor loading, the greater contribution of the corresponding food variables to the dietary pattern [52]. Dietary pattern scores were calculated by multiplying the factor loading by mean daily intake in grams of each food item and were categorised into quartiles, with quartile 4 indicating the highest degree of adherence [53,54]. If a food item was loaded on more than one factor, it was only retained in the PC with the highest factor loading unless the factor loading was in the opposite direction.

### 2.7. Ethical Considerations

Ethics approval for the AIM study was obtained from the South Eastern Sydney Local Health District Human Research Ethics Committee (2019/ETH11443). Participants provided written consent for data to be used in future studies. All data were analysed in a de-identified format.

### 2.8. Statistical Analysis

All statistical analyses were conducted in IBM^®^ SPSS^®^ Statistics (V29, IBM Australia Ltd., Sydney, Australia) [55]. Baseline characteristics were presented as median (interquartile range (IQR)) for continuous variables, and number of participants with percentages for categorical variables. The assumption of normality was assessed with Kolmogorov–Smirnov or Shapiro–Wilk tests. Differences in continuous variables were assessed with Mann-Whitney U test or Kruskal–Wallis test. Pairwise comparisons after Kruskal–Wallis test were conducted using Dunn’s test with Bonferroni adjustment. Differences in categorical variables were assessed with the Chi-squared test for independence or Fisher’s exact and adjusted standardised residuals. The association between PC factor score quartiles and categorical disease activity measures were tested using the Chi-square test. Adjusted standardised residuals and logistic binomial regression were conducted for the PC quartiles shown to have a statistically significant association with disease activities in the Chi-squared test of association. The Chi-squared test of association was performed to examine any association between FCP categories and PC factor score quartiles. A one-way ANOVA was conducted for each variable to test for significant differences between clusters. The Chi-squared test was performed to examine any association between clusters and clinical activity scores, FCP, and the composite classification, and for differences between clusters and proportion of UC vs. CD participants. The Spearman’s correlation was performed to examine correlations between dietary pattern scores within the IBD cohort.

## 3. Results

Seventy-seven participants were excluded due to having incomplete or missing diet data. No participants reported total energy <500 kcal; however, seven participants (1.7%) were excluded due to reported total energy >3500 kcal/day. A total of 412 participants were included in the analysis, consisting of 223 adults with IBD (117 with CD, 106 with UC) and 189 healthy adults. The median age was 45 years (IQR 34–56) and 56.3% were female (n = 232). Most (n = 292; 71%) participants had completed university-level education, and 314 (76%) reported having a post-school qualification. The median BMI was 24.4 (IQR 22–27.3) kg/m^2^. Over a third (37%) of participants reported a monthly income of AUD 1001–5000 from all income sources, equalling at or below the median Australian income in 2023 [56]. One-quarter (25%) reported a monthly income of AUD 5001–10,000 from all income sources. Participants with IBD were more likely to have completed secondary school as their highest level of education, and less likely to have completed university. The median partial Mayo at baseline was 1 (IQR 0–2) and the median CDAI at baseline was 92 (48–168), indicating clinical remission in UC and CD, respectively. The median BMI was higher in the CD and UC cohorts compared to the HC cohort. A higher proportion of participants with CD had ever smoked or were current smokers relative to the HC and UC cohorts. The HC cohort was more ethnically diverse than the IBD cohorts, with a higher proportion of participant from Southern, Eastern, and South-East Asian ethnicities and a lower proportion of Oceanian ethnicities. The remaining baseline characteristics were comparable between groups (Table 1). A higher proportion of participants in the CD cohort than the UC cohort had extra-intestinal manifestations, and were on higher-level medications including Thiopurines, Methotrexate, and advanced therapies (Appendix A).

### 3.1. Dietary Intake

There were no significant differences in nutrient or food group intake between the IBD vs. HC cohort. Compared to the CD cohort, the UC cohort had a higher median intake of anthocyanins (UC 10.56 [IQR 1.8–35.7] vs. CD 3.4 [0.5–3.4] mg/day; *p* = 0.007) and lignans (UC 0.2 [IQR 0.0–0.7] vs. CD 0.0 [0.0–0.2] mg/day; *p* = 0.006) (Table 2). All cohorts consumed an inadequate intake of food groups compared to the recommended intakes from the Australian Guide to Healthy Eating [30], except for the intake of meat/meat alternatives for female participants, as shown in Table 3. Intake of total fermented foods and probiotics was very low, with a median intake of 0.0 across all groups. No significant differences in nutrients, fermented foods, probiotics, or food groups were seen between IBD participants who were in clinical remission or who had a clinically active disease. See Appendix A for additional nutrient intake data for the HC and IBD cohorts. There were no significant differences in nutrient intake between cohorts.

### 3.2. Faecal Calprotectin

FCP data were available for 211 participants (HC = 100, CD = 49, UC = 62). There were 161 participants with no inflammation (FCP 0–50 μg/g, n = 65 with IBD); 22 with mild inflammation (FCP 51–150 μg/g, n = 19 with IBD); and 28 with high inflammation (FCP > 150 μg/g, n = 27 with IBD). The median FCP differed between IBD vs. HC (33.0 [IQR 0.0–147.0] vs. 0.0 [0.0–0.0] μg/g; *p* < 0.001) but not between CD vs. UC (24.0 [IQR 0.0–131.0] vs. 36.0 [0.0–167.8] μg/g, *p* = 0.380). Participants in the IBD cohort with no FCP inflammation had a significantly higher intake of total, insoluble, and soluble fibre than those with mild FCP inflammation (*p*-values 0.025, 0.011, and 0.043, respectively) (Table 4). However, there was no difference in fibre intake between no vs. high FCP inflammation and mild vs. high FCP inflammation.

### 3.3. Composite Classification of Disease Activity

There were 55 participants with IBD with both clinical and FCP remission (UC = 33, CD = 22) and 51 participants with either clinical and/or FCP active disease (UC = 27, CD = 24). In the UC cohort, intake of anthocyanins, flavones, phenolic acids, lignans, and total polyphenols was lower in participants in composite remission compared to those with active disease (Total polyphenols 205.9 mg [IQR 75.6–471.7] vs. 369.1 mg [219.7–624.7]; *p* 0.022. In the CD cohort, intake of anthocyanins was higher amongst participants in composite remission compared to those with active disease (6.1 mg [IQR 2.1–22.3] vs. 1.9 mg [1.1–12.8]; *p* 0.034)

### 3.4. Dietary Patterns

We identified seven distinct dietary patterns derived through PCA in the IBD cohort, explaining 54% of the total variance in dietary intake (see Appendix A for image of the scree plot to justify inclusion of seven dietary patterns). Eight food variables were excluded from the dietary pattern due to having a KMO test score of <0.5. Pattern 1, named ‘High plant diversity’, was characterised by higher intake of vegetables, fruit, and legumes. Pattern 2, named ‘Tea, coffee, and alcohol’, was characterised by higher intake of tea/coffee and alcohol. Pattern 3, named ‘Meat eaters’, was characterised by higher intake of red meat, pork, processed meat, and poultry. Pattern 4, named ‘Sweet discretionary’, was characterised by higher intake of sweet biscuits/cakes/pastries and chocolate/confectionary. Pattern 5, named ‘Animal fat’, was characterised by higher intake of butter, full-cream milk, and high-fibre cereals. Pattern 6, named ‘Cheese and yoghurt’, was characterised by higher intake of yoghurt and cheese/sour cream. Pattern 7, named ‘Vegan-style’, was characterised by higher intake of plant milk and nuts. These patterns explain 13.6%, 10.0%, 7.4%, 6.7%, 6.1%, 5.5%, and 4.6% of total variance in dietary intake, respectively, within the IBD cohort (Table 5).

We found five distinct dietary patterns derived through PCA in the HC cohort, accounting for 48% of the total variance in dietary intake. Eleven food variables were excluded from the dietary pattern due to having a KMO test score of <0.5. Pattern 1, named ‘High plant diversity’, was characterised by higher intake of vegetables, fruit, nuts, and legumes. Pattern 2, named ‘Sweet discretionary’, was characterised by higher intake of sweet biscuits/cakes/pastries, chocolate/confectionary, ice-cream, and sugary beverages. Pattern 3, named ‘Meat eaters’, was characterised by higher intake of poultry, pork, red meat, and fish/seafood. Pattern 4, named ‘Animal fat’, was characterised by higher intake of cheese/sour cream, yoghurt, sauces/condiments, and processed meats. Pattern 5, named ‘Pescetarian-style’, was characterised by higher intake of olive oil, legumes, nuts, and fish/seafood. These patterns explained 16.7%, 10.6%, 8.0%, 6.9%, and 6.5% of variance, respectively (Table 6).

To explore relationships between adherence to different PCA dietary patterns in the IBD cohort, we performed a correlation matrix of the dietary pattern scores for each of the seven dietary patterns (see Figure 2). There was a weak positive correlation between the ‘High plant diversity’, ‘Cheese and yoghurt’, and ‘Vegan-style’ patterns, and a weak negative correlation between the ‘High plant diversity’ pattern and the ‘Meat eaters’ pattern. Additionally, there was a strong negative correlation between the ‘Vegan-style’ pattern and the ‘Meat eaters’ pattern. There was a weak positive correlation between the ‘Tea, coffee, and alcohol’, ‘Sweet discretionary’, and ‘Animal fat’ dietary patterns.

### 3.5. Relationship Between Dietary Patterns and Disease Activity

In the combined IBD cohort, there was no significant relationship between adherence to any dietary pattern and clinical disease activity, categorised as remission vs. active. In the CD cohort, there was a difference in the median CDAI score between different quartiles of adherence to the ‘High plant diversity’ dietary pattern score, with the highest median CDAI in quartile 3 (159.3 [IQR 84.9–204.0]) compared to quartile 1 (89.5 [49.0–138.1]), quartile 2 (33 [13.6–92.6]), and quartile 4 (112.9 [56.5–166.7]) (*p* 0.004) (see Figure 3). Amongst the CD cohort, adjusted residuals identified high adherence to the ‘Meat eaters’ dietary pattern was associated with mild intestinal inflammation (50–150 μg/g). In the total IBD cohort, the degree of adherence to the ‘Vegan-style’ dietary pattern score was directly proportional to the FCP category, with low adherence associated with no inflammation and moderate-high adherence associated with high intestinal inflammation (*p* 0.015). There was no significant difference in median fibre intake between quartiles of adherence to the ‘High plant diversity’ or ‘Meat eaters’ dietary pattern in the CD cohort, nor between quartiles of adherence to the ‘Vegan-style’ dietary pattern in the IBD cohort. There was no association between dietary patterns and FCP in either the UC or HC cohorts. There was no difference in the proportion of IBD participants in remission vs. active disease (as defined by the composite classification) between the PCA dietary patterns.

We additionally performed cluster analysis to further explore dietary patterns in this cohort. The methods and full results are reported in Appendix A. Cluster analysis separates participants into mutually exclusive groups, called clusters, based on similarity of their food consumption habits. Briefly, we identified three dietary clusters for adults with IBD and three dietary clusters for HC adults. There were no significant differences in the proportion of participants with CD or UC in the IBD dietary clusters, and no significant differences in the proportion of IBD participants with clinical disease activity or remission in the clusters. There was no difference in median FCP, proportion of participants in each of the FCP categories, or proportion of participants in composite classification categories between dietary clusters.

## 4. Discussion

A major aim of this study was to investigate whether the dietary intake of Australian adults with IBD differed to that of healthy controls (HCs), and whether dietary patterns were associated with disease activity. We found a significantly higher intake of anthocyanins and lignans in adults with UC compared to CD; however, these differences were minimal and unlikely to contribute to differences in clinical outcomes. We found minimal differences in food group and nutrient intake between the combined IBD cohort and HCs. High adherence to a ‘High plant diversity’ and ‘Meat eaters’ dietary patterns within the CD cohort was associated with increased CDAI and FCP disease activity markers, respectively. Additionally, we found higher adherence to a ‘Vegan-style’ dietary pattern was associated with higher FCP in the combined IBD cohort.

A key finding from this study is that all participants, whether healthy controls or those with IBD, consumed poor quality diets with inadequate servings of most food groups compared to Australian reference standards [30]. Previous literature amongst IBD cohorts globally have similarly demonstrated low adherence to dietary guidelines. Lambert et al. (2021) [57] examined dietary intake amongst adults with IBD from multiple countries and found that all cohorts reported inadequate intake of legumes, fruits, vegetables, and dairy. Similar results were seen amongst a study of 100 Australian adults with IBD [58]. Studies from the broader Australian population consistently show that most individuals of diverse ages do not meet the dietary guideline recommendations [59]. In contrast to previous studies, the present analysis found no differences in adherence to the Australian Dietary Guidelines between adults with IBD compared to healthy participants. Peters et al. (2020) [60] and Opstelten et al. (2019) [61] compared dietary intake amongst Dutch adults with IBD to healthy controls and found that adults with IBD consumed lower amounts of dairy, vegetables, nuts, cereals, and grains, and higher intake of meats assessed by an FFQ. Given that adults with IBD have reported high rates of food avoidance, restrictive eating, and low food-related quality of life [62], the finding of comparable dietary intake to HCs in the present study was unexpected. Another unexpected finding was the low use of probiotic supplements in the IBD cohort and no difference in probiotic intake between the IBD and HC cohorts. This contrasts with previous literature which has demonstrated that adults with IBD have significantly higher rates of probiotic use than healthy controls [63]. The low reported probiotic intake in this cohort may be due to the way these data were collected. Participants were asked in the demographic questionnaire whether they currently take any medications not prescribed by their doctor, including vitamins, herbal, or complementary medicine. It is possible that many participants omitted probiotics in their answer to this question.

The low intake of grains, vegetables, and fruits in this cohort has implications for adequate consumption of dietary fibre and microbiota-accessible carbohydrates. Dietary fibres are carbohydrates that are resistant to digestion in the upper parts of the gastrointestinal tract and are usually partially or completely fermented in the distal small intestine or colon. Many types of soluble dietary fibre and resistant starch are microbiota-accessible carbohydrates and are fermented by gut bacteria to produce short chain fatty acids [64]. The IBD cohort in this study had a medium soluble fibre intake of 4.98 g/day, comparable to previous dietary intake studies in adults with IBD [16,65,66]. This intake is substantially lower than the dose of 15 g/day soluble fibre intake supplemented in a clinical trial with 103 adults with CD and supplemented in another clinical trial with 31 adults with UC [67,68]. Both studies showed a significant improvement in clinical disease scores after 4 weeks and 9 weeks, respectively. The reported median resistant starch intake within the current IBD cohort of 1.71 g/day is substantially lower than the recommendation of 20 g resistant starch/day for healthy adults [69]. Clinical trials investigating the effects of resistant starch in IBD are lacking, but pre-clinical evidence suggests that resistant starch may ameliorate disease in murine models of colitis [70,71]. The association between higher dietary fibre intake and lower FCP inflammation in the IBD cohort in the current study may indicate that participants with lower inflammation and therefore controlled disease are consuming more dietary fibre, and alternatively that participants with higher inflammation are consuming less dietary fibre. This association may be the result of participants with active IBD being recommended to reduce their fibre intake by health professionals, despite evidence suggesting the effectiveness of widespread low-fibre diets during a disease flare is sparse [4,72]. Participants in the IBD cohort may also be reducing dietary fibre intake of their own accord due to a lack of tolerance during a disease flare [73,74]. Nevertheless, given the inverse association between dietary fibre and inflammation, and the overall inadequate consumption of dietary fibre compared to recommended intakes, the findings from the current study suggest there is the opportunity for beneficial dietary modification amongst Australian adults with IBD to improve dietary fibre intake. People living with IBD should be recommended to consume healthy eating patterns when not in an acute disease flare, including national dietary guidelines or the Mediterranean diet, both of which emphasise a higher dietary fibre intake than consumed by the current cohort [30,75].

The dietary pattern analysis revealed substantial variation in dietary intake within both the HC and IBD cohorts. PCA was utilised to condense the dietary data into consumption patterns, highlighting the primary ways in which the 28 included food variables are typically consumed together. Seven distinct dietary patterns were identified, accounting for 54% of the total variance in habitual dietary intake. Additional dietary patterns, representing the remaining 46% of variance, were not captured by the retained components in this PCA analysis. The surprising positive association between the ‘Vegan-style’ dietary pattern and FCP in the combined IBD cohort, and the ‘High plant diversity’ dietary pattern and CDAI in the CD cohort contrasts with previous dietary pattern analyses in IBD cohorts. A retrospective study amongst 691 adults with IBD found that following a plant-based dietary pattern (high intake of fruits, vegetables, and grains) compared to a Western dietary pattern (low intake of fruits, vegetables, plant proteins, and cooked grains) for the preceding 3 months was associated with a lower odds of active symptoms at the time of the survey for CD, UC, and combined IBD cohorts [7]. That study reported the proportion of participants in clinical remission was highest amongst those who adhered to plant-based dietary patterns [7]. The association between high adherence to the ‘Meat eaters’ pattern and mild intestinal inflammation in the current study is comparable with a previous study amongst Dutch adults, in which a PCA-derived dietary pattern characterised by a high intake of grains, oils, potatoes, processed meat, red meat, sweet foods, and sauces/condiments was significantly associated with an increased risk of developing a clinical disease flare during a follow-up period of up to 2 years [8]. A strength of these two previous dietary pattern studies was the ability to characterise dietary patterns before the onset of symptoms, therefore inferring temporality.

A key finding from the exploratory dietary pattern analysis is that participants with IBD in this study have diverse patterns of eating and that multiple patterns of eating are associated with increased disease activity. The correlation analysis revealed that high adherence to the ‘High plant diversity’ and ‘Vegan-style’ patterns correlated with lower adherence to the ‘Meat eaters’ pattern. Although consumption of these dietary patterns is not mutually exclusive due to the nature of PCA analysis, the correlations indicate that some participants with IBD in this study are consuming dietary patterns with more fruits, vegetables, and plant-based alternatives, and others are consuming dietary patterns characterised by higher meat consumption. Interestingly, both are associated with disease activity. Given the cross-sectional nature of this study, we are unable to consider temporality or causality, and therefore unable to determine whether changes to disease activity precede changes to dietary intake, or whether changes to dietary intake precede changes to disease activity. Around 57% of people with IBD believe that food can contribute to relapse or disease management [76,77], and rates of dietary modification are shown to be higher amongst participants with active disease compared to those in remission [78]. It is possible that, within this IBD cohort, some individuals with active disease were modifying their diet to try to manage their disease and reduce symptoms, contributing to the association between healthier dietary patterns and active disease. Further longitudinal studies that measure dietary patterns and their association with clinical disease activity, FCP, and gastrointestinal symptoms over time will be useful to expand upon the findings of the current study. It is important to consider that within the current IBD cohort, it is likely that many participants have not received formal advice from a nutritional professional on dietary management of IBD. A 2016 audit of public IBD services in Australia found that only 56% of Crohn’s Disease patients and 44% of Ulcerative Colitis patients who were recommended a nutritional supplement during a hospital admission were followed up by a dietitian after discharge. Only 1% of sites included in the audit had a full-time IBD dietitian [79]. Data on the proportion of Australians with IBD who have accessed private dietitian services for dietary therapy remain limited. Participants with IBD in this cohort may not have modified their diet for health or disease management, and therefore the diverse dietary patterns seen may reflect diversity in dietary intake across all human populations. The diversity in dietary patterns and poor adherence to dietary guidelines evident in this cohort emphasise the need for individualised dietetic therapy to improve dietary intake and reduce chronic disease risk amongst Australian adults with IBD.

There are several limitations to the current study. First, a limitation of a posteriori dietary patterns is the number of researcher-determined decisions involved in collapsing dietary variables into fewer groups, assigning labels, and determining patterns to retain [47]. Second, the percentage of total energy (%TE) was not considered in the dietary pattern analyses, and food variables were analysed in the form of grams per day. This decision ensured the dietary patterns captured the contribution of low energy-contributing foods, particularly tea and coffee, due to their high polyphenol content. Additionally, non-energy-adjusted analyses have demonstrated higher sensitivity for detecting the contribution of low-energy foods including fruits and vegetables [80]. Approximately half the participants in this study were in clinical remission. This may represent a degree of selection bias, with participants who are relatively healthy being more likely to participate in research than those with severe disease. Therefore, the findings of this study are generalizable only to cohorts in remission or with mild to moderate IBD. Finally, given the cross-sectional nature of this study, we are unable to determine temporality or causality in the observed associations between dietary intake and disease outcomes. Longitudinal and clinical studies are needed to expand upon the limitations of this study design.

A strength of this study is that dietary data for the nutrient intake calculations were assessed via 3-day food records, which are an accurate and valid method of assessing dietary intake [81]. To our knowledge, this study is the first to calculate the habitual intake of dietary polyphenols and sub-categories of dietary fibres, as opposed to only total dietary fibre. The inclusion of two clinically relevant measures of disease activity is a unique strength of the present study and allowed for a detailed analysis of the relationship between diet and disease activity. Furthermore, this is the first a posteriori dietary pattern study in IBD to include FCP data as an outcome, and the first to examine disease activity as a composite of both FCP and clinical disease status.

## 5. Conclusions

The findings from this study suggest that Australian adults with IBD do not meet recommended dietary intakes of food groups and fibres, but that this lack of adherence to recommended dietary guidelines is comparable to that of healthy Australians. Habitual dietary fibre intake was inadequate compared to Australian recommended intakes, but higher dietary fibre intake was associated with no intestinal inflammation. Principal component analysis identified that dietary patterns characterised either by a high intake of plant foods or meat products were both positively associated with indicators of active IBD. It is possible that, within this IBD cohort, some individuals with active disease were modifying their diet to try to manage their disease and reduce symptoms, contributing to the association between healthier dietary patterns and active disease. Further longitudinal studies that measure dietary patterns and their association with disease activity, inflammation, and gastrointestinal symptoms over time are necessary to establish temporality and causality in this association. Such studies will be useful to determine whether changes to dietary intake precede changes to disease activity, or whether participants are changing their dietary intake in response to changes in disease activity. It is likely the findings of this study are generalizable only to cohorts in remission or with mild to moderate IBD. The diversity in dietary pattern consumption and inadequate intake of core food groups and dietary fibre emphasise the need for individualised dietetic therapy amongst Australian adults with IBD, to facilitate beneficial dietary modification.

## Figures and Tables

**Figure 1 nutrients-16-04349-f001:**
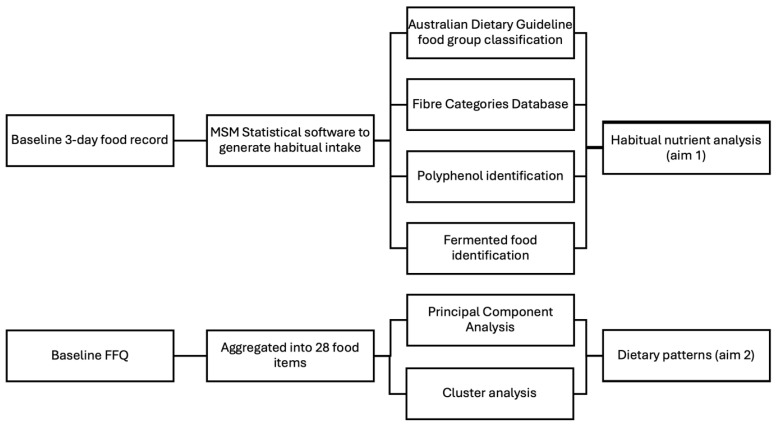
Summary of steps undertaken to transform FFQ and 3-day food record dietary data.

**Figure 2 nutrients-16-04349-f002:**
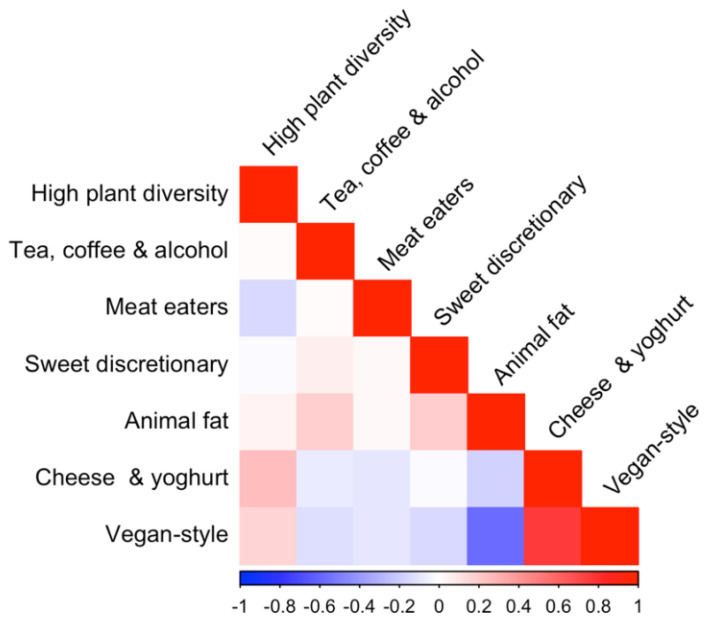
Correlation matrix of adherence to the seven PCA-derived dietary patterns in the IBD cohort.

**Figure 3 nutrients-16-04349-f003:**
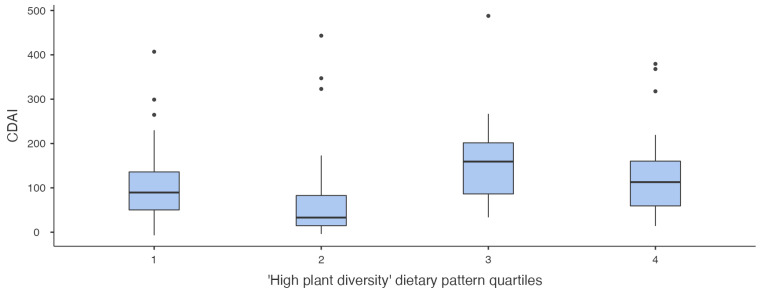
Boxplot of the CDAI score for each quartile of the ‘High plant diversity’ dietary pattern amongst the CD cohort.

**Table 1 nutrients-16-04349-t001:** Population characteristics for all groups.

Characteristic	HC (n = 189)	CD (n = 117)	UC (n = 106)	*p*-Value
Age in years, median (IQR)	42.0 (32.0–53.0)	47.0 (35.3–58.8)	46.0 (36.0–58.0)	0.072 ^†^
BMI kg/m^2^, median (IQR)	23.6 (21.50–26.0)	25.1 (22.1–28.3)	25.6 (22.5–28.30)	<0.001 *^†^
Sex, n female (%)	105 (55.6)	65 (55.5%)	62 (58.5)	0.871 ^‡^
Post-school qualifications, n (%)	151 (81.6%)	83 (70.9%)	80 (75.5%)	0.091 ^‡^
Completed university-level education, n (%)	161 (85.6%)	66 (56.4%)	65 (61.9%)	<0.001 *^‡^
Ever smoked, n (%)	30/145 (20.7%)	40/117 (34.2%)	41/106 (38.7%)	0.005 *^‡^
Current smoker, n (%)	5/145 (3.5%)	14/117 (12%)	4/104 (3.8%)	0.016 *^§^
FCP, median (IQR)	0.0 (0.0–0.0) μg/g	24.0 (0.0–131.0) μg/g	36.0 (0.0–167.8) μg/g	<0.001 *^†^
Income, n (%)				0.298 ^‡^
AUD 0–1000	19 (10.5%)	11 (10.1%)	7 (7.3%)	
AUD 1001–5000	57 (31.5%)	55 (50.5%)	41 (42.7%)	
AUD 5001–10,000	52 (28.7%)	23 (21.1%)	31 (32.3%)	
AUD 10,001–20,000	18 (9.9%)	10 (9.2%)	6 (6.3%)	
>AUD 20,001	35 (19.3%)	10 (9.2%)	11 (11.5%)	
Ethnicity, n (%)				
Oceanian	76 (43.2%)	66 (61.1%)	58 (58.0%)	0.012 ^‡^
European	35 (19.9%)	26 (24.1%)	27 (27.0%)	
North-East Asian	21 (11.9%)	3 (2.8%)	0	
Southern, South-Eastern and Eastern Asian	25 (26.1%)	5 (4.6%)	4 (4.0%)	
North and South American	4 (2.3%)	1 (0.9%)	1 (1.0%)	
North African and Middle Eastern	10 (5.7%)	3 (2.8%)	5 (5.0%)	
Sub-Saharan African	1 (0.6%)	1 (0.9%)	1 (1.0%)	
Other	4 (2.3%)	2 (1.9%)	4 (4.0%)	

^†^ *p*-values from the Kruskal–Wallis test of between-group differences. ^‡^ *p*-values from Chi-squared test of between-group differences. ^§^ *p* value from Fisher’s exact test of between-group differences * *p*-value < 0.05, indicating statistical significance. HC = healthy controls; CD = Crohn’s Disease; UC = Ulcerative Colitis; IQR = interquartile range; BMI = body mass index; FCP = faecal calprotectin.

**Table 2 nutrients-16-04349-t002:** Descriptive statistics for food and nutrient intake from total sample (n = 412) from 3-day food records.

Food and Nutrients Reported as Median (IQR)	IBD (n = 412)	HC (n = 189)	*p*-Value HC vs. IBD ^†^	CD (n = 117)	UC (n = 106)	*p*-Value CD vs. UC ^†^	IBD Clinical Remission (n = 142)	IBD Clinically Active Disease (n = 130)	*p*-Value Active vs. Remission ^†^	IBD Remission by Composite Classification (n = 55)	IBD Active Disease by Composite Classification (n = 51)	*p*-Value Active vs. Remission ^†^
Total dietary fibre (g/day)	17.0 (13.1–21.8)	16.7 (12.7–23.3)	0.510	16.0 (12.8–22.2)	17.5 (13.2–21.8)	0.757	16.8 (13.1–21.0)	17.6 (12.8–21.2)	0.751	17.7 (14.3–20.6)	17.2 (12.9–23.6)	0.770
Insoluble fibre (g/day)	11.6 (8.7–14.7)	12.0 (8.8–16.7)	0.289	11.5 (8.6–14.8)	11.8 (9.0–14.6)	0.705	11.6 (8.6–14.5)	12.0 (8.8–15.9)	0.728	12.2 (9.5–14.6)	12.0 (8.6–17.2)	0.808
Soluble fibre (g/day)	5.0 (3.6–6.5)	5.0 (3.5–6.3)	0.892	4.8 (3.5–6.8)	5.1 (3.7–6.4)	0.710	4.8 (3.5–6.5)	5.0 (3.7–6.6)	0.801	5.1 (3.4–6.7)	5.1 (3.4–6.7)	0.692
Resistant starch (g/day)	1.7 (1.2–2.4)	1.6 (1.2–2.2)	0.291	1.5 (1.0–2.3)	1.8 (1.3–2.5)	0.063	1.7 (1.2–2.4)	1.9 (1.1–2.5)	0.760	1.9 (1.3–2.6)	1.9 (1.3–2.6)	0.497
Anthocyanins (mg/day)	4.2 (1.6–21.5)	4.0 (1.7–28.8)	0.899	3.4 0.5–3.4)	10.6 (1.8–35.7)	0.007 *	4.6 (1.6–21.5)	3.7 (1.3–20.8)	0.747	5.7 (1.9–21.1)	3.8 (1.3–21.7)	0.782
Flavan-3-ols (mg/day)	27.3 (8.2–115.4)	20.9 (6.78–95.4)	0.731	25.3 (7.3–153.6)	29.7 (8.4–108.0)	0.685	25.2 (8.0–115.7)	30.6 (9.0–149.4)	0.962	24.5 (9.6–159.6)	35.5 (10.7–205.5)	0.393
Flavanones (mg/day)	1.4 (0.4–5.0)	1.3 (0.3–4.2)	0.815	1.1 (0.4–5.3)	1.6 (0.4–4.8)	0.568	1.4 (0.3–5.9)	1.3 (0.4–3.9)	0.548	1.6 (0.3–5.9)	0.9 (0.4–4.6)	0.839
Flavonols (mg/day)	7.6 (3.7–13.87)	8.1 (4.1–12.9)	0.781	7.2 (0.1–1.0)	8.1 (4.4–13.8)	0.320	7.6 (3.5–13.8)	8.5 (4.2–14.8)	0.706	7.1 (4.1–14.5)	10.0 (4.2–15.5)	0.402
Flavones (mg/day)	0.4 (0.2–1.0)	0.4 (0.1–0.9)	0.705	0.3 (3.1–14.5)	0.5 (0.2–1.0)	0.369	0.3 (0.1–0.9)	0.5 (0.2–1.0)	0.357	0.35 (0.2–1.0)	0.49 (0.1–1.1)	0.066
Isoflavones (mg/day)	0.0 (0.0–0.0)	0.0 (0.0–0.0)	0.215	0.0 (0.0–0.0)	0.0 (0.0–0.0)	0.921	0.0 (0.0–0.0)	0.0 (0.0–0.0)	0.686	0.0 (0.0–0.0)	0.0 (0.0–0.0)	0.786
Total flavonoids (mg/day)	57.1 (20.0–145.8)	60.1 (21.2–154.0)	0.908	49.0 (17.9–143.1)	69.1 (26.2–151.2)	0.223	58.5 (19.0–148.1)	56.9 (22.2–148.8)	0.840	62.9 (24.5–141.6)	109.4 (21.8–212.6)	0.192
Total phenolic acids (mg/day)	150.0 (51.0–300.7)	159.0 (44.3–348.9)	0.805	127.9 (43.9–291.8)	182.5 (62.8–338.1)	0.078	152.0 (46.8–332.5)	145.7 (53.0–294.9)	0.649	147.3 (43.9–334.1)	192.5 (71.4–297.6)	0.058
Stilbenes (mg/day)	0.0 (0.0–0.0)	0.0 (0.0–0.0)	0.422	0.0 (0.0–0.0)	0.0 (0.0–0.0)	0.864	0.0 (0.0–0.0)	0.0 (0.0–0.0)	0.675	0.0 (0.0–0.0)	0.0 (0.0–0.0)	0.981
Lignans (mg/day)	0.1 (0.0–0.34	0.1 (0.0–0.5)	0.757	0.0 (0.0–0.2)	0.1 (0.0–0.7)	0.006 *	0.1 (0.0–0.3)	0.1 (0.0–0.3)	0.593	0.1 (0.0–0.5)	0.0 (0.0–0.2)	0.636
Total polyphenols (mg/day)	267.0 (100.7–460.9)	280.7 (90.9–492.8)	0.680	248.7 (94.7–420.3)	271.6 (123.8–524.2)	0.159	271.6 (112.6–524.2)	246.1 (96.5–423.9)	0.411	252.9 (112.3–436.9)	301.8 (148.4–478.7)	0.469

^†^ *p*-values from the Mann–Whitney U test of statistical difference. * *p*-value < 0.05, indicating statistical significance. IQR = interquartile range; IBD = Inflammatory Bowel Disease; HC = healthy controls; CD = Crohn’s Disease; UC = Ulcerative Colitis.

**Table 3 nutrients-16-04349-t003:** Descriptive statistics for food group intake (n = 412) from 3-day food record.

Food Group in Servings/Day, Reported as Median (IQR)	IBD (n = 412)	HC (n = 189)	*p*-Value HC vs. IBD ^†^	CD (n = 117)	UC (n = 106)	*p*-Value CD vs. UC ^†^	IBD Clinical Remission (n = 142)	IBD Clinically Active Disease (n = 130)	*p*-Value Active vs. Remission ^†^	IBD Remission by Composite Classification (n = 55)	IBD Active Disease by Composite Classification (n = 51)	*p*-Value Active vs. Remission ^†^	ADG Recommended Serves M = Males F = Females
Grains	5.0 (3.8–6.5)	5.2 (4.1–6.5)	0.399	5.0 (3.86–6.2)	5.2 (4.1–6.6)	0.424	5.0 (3.8–6.6)	5.1 (4.0–6.0)	0.993	5.1 (3.9–6.6)	5.4 (4.0–6.4)	0.431	6 (M and F)
Wholegrains	1.2 (0.5–2.2)	1.4 (0.7–2.3)	0.247	1.2 (0.7–2.3)	1.2 (0.4–2.1)	0.261	1.2 (0.5–2.2)	1.2 (0.6–1.8)	0.696	1.2 (0.6–2.5)	1.0 (0.5–1.8)	0.109	
Refined grains	3.5 (2.3–4.6)	3.4 (2.2–4.5)	0.714	3.4 (2.3–4.4)	3.6 (2.3–4.8)	0.310	3.5 (2.3–4.4)	3.5 (2.4–4.7)	0.874	3.5 (2.7–4.5)	3.6 (2.6–5.1)	0.676	
Vegetables	3.4 (2.4–4.3)	3.4 (2.4–4.4)	0.400	3.5 (2.3–4.5)	3.3 (2.5–4.2)	0.627	3.4 (2.4–4.3)	3.6 (2.3–4.5)	0.508	3.4 (2.4–4.5)	3.8 (2.5–5.0)	0.350	6 (M), 5 (F)
Fruit	1.0 (0.5–1.7)	0.9 (0.4–1.8)	0.587	1.0 (0.4–1.8)	0.9 (0.5–1.7)	0.815	0.9 (0.4–1.8)	1.1 (0.6–1.6)	0.413	1.0 (0.5–2.2)	1.1 (0.5–2.1)	0.390	2 (M and F)
Dairy	1.5 (1.0–2.2)	1.5 (0.8–2.2)	0.607	1.5 (1.0–2.0)	1.7 (1.1–2.3)	0.160	1.5 (1.0–2.1)	1.6 (1.1–2.2)	0.463	1.5 (1.0–1.9)	1.7 (1.0–2.4)	0.329	2.5 (M and F)
Meat/meat alternatives	2.6 (1.9–3.2)	2.4 (1.7–3.3)	0.241	2.7 (1.9–3.5)	2.6 (1.9–3.1)	0.383	2.7 (1.9–3.3)	2.4 (1.9–3.0)	0.474	2.5 (1.8–3.2)	2.4 (1.8–3.0)	0.314	3 (M), 2.5 (F)

^†^ *p*-values from the Mann–Whitney U test of statistical difference. IQR = interquartile range; IBD = Inflammatory Bowel Disease; HC = healthy controls; CD = Crohn’s Disease; UC = Ulcerative Colitis; ADG = Australian Dietary Guidelines.

**Table 4 nutrients-16-04349-t004:** Median (IQR) dietary fibre intake for each category of FCP in the IBD cohort.

FCP Category	No Inflammation, n = 65 (0–50 μg/g)	Mild Inflammation, n = 19 (50–150 μg/g)	High Inflammation, n = 27 (>150 μg/g)	*p*-Value Between-Gro-up Difference ^†^	*p*-Value Pair-Wise Comparison of No vs. Mild Inflammation ^‡^
Total dietary fibre (g/day)	17.72 (15.02–22.55)	14.62 (7.60–18.54))	15.83 (13.15–20.26)	0.023 *	0.025 *
Insoluble fibre (g/day)	12.45 (10.61–15.55)	9.91 (5.80–13.73)	11.33 (8.65–14.19)	0.010 *	0.011 *
Soluble fibre (g/day)	5.224 (4.00–6.86)	4.35 (2.02–5.55)	5.27 (3.41–6.46)	0.050	0.043 *

^†^ *p*-values from the Kruskal–Wallis test of statistical difference. ^‡^ *p*-values from the Dunn’s test with Bonferroni adjustment. * *p*-value < 0.05, indicating statistical significance. IQR = interquartile range; IBD = Inflammatory Bowel Disease; FCP = faecal calprotectin.

**Table 5 nutrients-16-04349-t005:** Factor loadings for food groups that load (|>0.30|) in each PCA-derived dietary pattern for the IBD cohort.

Food Variable ^†^	High Plant Diversity	Tea, Coffee, and Alcohol	Meat Eaters	Sweet Discretionary	Animal Fat	Cheese and Yoghurt	Vegan-Style
Eigenvalue	3.42	2.509	1.859	1.683	1.507	1.356	1.153
% variance explained	13.685	10.036	7.437	6.732	6.028	5.425	4.613
All other veg	0.795						
Orange/red veg	0.774						
Green veg	0.760						
Starchy veg	0.540						
Legumes	0.451						
Other fresh fruit	0.451						
Alcoholic beverages		0.937					
Tea/coffee		0.934					
Red meat			0.769				
Pork			0.662	−0.309			
Processed meats			0.642				
Poultry			0.574				
Sweet biscuits/pastries				0.793			
Chocolate/lollies				0.726			
Butter					0.729		
Full-cream milk					0.502		−0.405
High-fibre cereals					0.498		
Yoghurt						0.806	
Cheese/sour cream						0.629	
Plant milk							0.897
Nuts							0.448
Dairy desserts/ice-cream							
Berries	0.319						
Diet soft drink/water					−0.417		

^†^ Only food variables with factor loadings ≥|0.30| are displayed and listed in order for ease of interpretation. PCA = principal component analysis; IBD = Inflammatory Bowel Disease.

**Table 6 nutrients-16-04349-t006:** Factor loadings for food groups that load highly (|>0.30|) in each PCA-derived dietary pattern for the HC cohort.

Food Variable ^†^	High Plant Diversity	Sweet Discretionary	Meat Eaters	Animal Fat	Pescetarian-Style
Eigenvalue	3.851	2.442	1.844	1.580	1.484
% variance explained	16.744	10.619	8.018	6.869	6.452
All other veg	0.853				
Orange/red veg	0.824				
Green veg	0.771				
Berries	0.737				
Sweet biscuits/pastries		0.778			
Chocolate/lollies		0.754			
Dairy dessert/ice-cream		0.731			
Sugary beverage		0.438		−0.415	
Poultry			0.805		
Pork			0.786		
Red meat			0.431		−0.369
Cheese/sour cream				0.770	
Yoghurt				0.565	
Sauces/condiments				0.485	
Processed meats				0.472	
Olive oil					0.698
Legumes	0.389				0.585
Nuts	0.354				0.524
Seafood			0.324		0.368
High-fibre cereals					
Other fresh fruit	0.408				

^†^ Only food variables with factor loadings ≥ |0.30| are displayed and listed in order for ease of interpretation. PCA = principal component analysis; HC = healthy control.

## Data Availability

Data are available on request.

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
