# Peer review of "Dietary Patterns and Fibre Intake Are Associated with Disease Activity in Australian Adults with Inflammatory Bowel Disease: An Exploratory Dietary Pattern Analysis [Author-notes fn1-nutrients-16-04349]"

_nutrients, 2024, doi:10.3390/nu16244349_

Round 1
Reviewer 1 Report
Comments and Suggestions for Authors
Thank you for the opportunity to review this interesting article related to dietary patterns and IBD activity. This appears to be a thorough and well conducted study that adds to a body of knowledge. Overall the article is well presented with key data included within the main article. Supplementary files are available.
My only comment is that I think there are a number of formatting and/or typing errors in the article that I have highlighted in yellow on the attached document. The ethical considerations section is missing completely. These need attention in order to complete the work
Thank you

Author Response
Comments: [Thank you for the opportunity to review this interesting article related to dietary patterns and IBD activity. This appears to be a thorough and well conducted study that adds to a body of knowledge. Overall the article is well presented with key data included within the main article. Supplementary files are available.
My only comment is that I think there are a number of formatting and/or typing errors in the article that I have highlighted in yellow on the attached document. The ethical considerations section is missing completely. These need attention in order to complete the work]
Response: [Thank you kindly for your review of this study and for identifying the formatting errors. See these errors corrected in the attached manuscript, as well as the completed ethical considerations section. The changes are highlighted. ]

Reviewer 2 Report
Comments and Suggestions for Authors
Dear authors and editor,
The manuscript titled "Dietary patterns and fibre intake are associated with disease activity in Australian adults with Inflammatory Bowel Disease" explores the relationship between dietary intake and inflammatory bowel disease (IBD) activity. Below, I present comments and suggestions for improving the manuscript.
Abstract
1. Include the study design in the title.
2. The abstract is concise and provides a clear overview of the study objectives, methodology, and key findings.
Consider explicitly highlighting the unique contributions of this study, such as the inclusion of faecal calprotectin (FCP) as a measure of inflammation and the use of dietary pattern analysis.
Introduction
3. The introduction is well-structured and provides a strong rationale for the study. However, the discussion of dietary patterns could be expanded to include global comparisons or references to similar studies outside Australia.
4. The rationale for the three specific aims is clear, but linking them to gaps in the literature would strengthen the argument.
Materials and Methods
5.The description of dietary data collection and analysis is comprehensive and aligns with standard methodologies. The use of PCA and cluster analysis is appropriate for dietary pattern identification.
6.Ethical considerations and participant inclusion/exclusion criteria are clearly stated.
7.The justification for categorising faecal calprotectin levels is robust, but additional detail on how missing data were managed would be useful.
8.How was the sample size calculated?
Results
9.The results are well-presented, with tables and figures enhancing clarity. However, consider reorganising results related to dietary patterns and disease activity for better flow.
10.It is recommended to include table abbreviations at the bottom of the tables.
Discussion
11.The discussion appropriately contextualises findings within the broader literature. However, the unexpected association between plant-based diets and increased disease activity requires further exploration.
12.Consider discussing the implications of low dietary fibre intake and its potential role in IBD management more extensively.
13.The limitations section is well-developed but could be strengthened by discussing the cross-sectional design's inability to establish causality.
Conclusion
14.The conclusions align with the study objectives and findings. They effectively summarise the study's relevance but could include a stronger call for future longitudinal studies to establish causal links.
References
15.The references are comprehensive and formatted according to the journal's guidelines.
Author Response
Dear Reviewer,
Thank you kindly for reviewing this paper and for providing the suggestions for improvement. See below responses to each suggestion:
Comment 1: [Include the study design in the title]
Response 1: The title has been modified to include the study design. The new title is "Dietary patterns and fibre intake are associated with disease activity in Australian adults with Inflammatory Bowel Disease: an exploratory dietary pattern analysis"
Comment 2: [The abstract is concise and provides a clear overview of the study objectives, methodology, and key findings.
Consider explicitly highlighting the unique contributions of this study, such as the inclusion of faecal calprotectin (FCP) as a measure of inflammation and the use of dietary pattern analysis]
Response 2: Thank you for suggesting we highlight the uniqueness of the study in the abstract. We have included a sentence to highlight this at Line 45 of the abstract.
Comment 3: [The introduction is well-structured and provides a strong rationale for the study. However, the discussion of dietary patterns could be expanded to include global comparisons or references to similar studies outside Australia.]
Response 3: Thank you for this comment. We have discussed a study conducted amongst Indian patients with IBD at line 84 to expand the discussion to global cohorts. The three referenced dietary pattern studies on line 67 were conducted amongst Dutch, American and Brazilian.
Comment 4: [The rationale for the three specific aims is clear, but linking them to gaps in the literature would strengthen the argument.]
Response 4: Thank you for this comment. We have added a sentence at Line 98 to summarise the gaps in the literature identified, and therefore link the aims of the study to these gaps.
Comment 5: [The justification for categorising faecal calprotectin levels is robust, but additional detail on how missing data were managed would be useful]
Response 5: There was no missing FCP data in this study. FCP was available for 211 participants included in this study, and all 211 samples were analysed. This has been clarified at Line 184.
Comment 6: [How was the sample size calculated?]
Response 6: The sample size included all adult participants who submitted dietary records between 1 July 2019 to 30 December 2022. This is mentioned at Line 124 of the methods section.
Comment 7: [The results are well-presented, with tables and figures enhancing clarity. However, consider reorganising results related to dietary patterns and disease activity for better flow].
Response 7: Thank you for this comment. We decided to organise the results of this study presenting the dietary intake data and relationship to disease activity first, and then followed by the results of the dietary pattern analysis and relationship with disease activity second. This is in line with the order layed out in the aims. We are not sure how to reorganise the results to further improve the clarity or flow of the study.
Comment 8: [It is recommended to include table abbreviations at the bottom of the tables]
Response 8: Thank you for this suggestion. We have included table abbreviations at the bottom of each included table.
Comment 9: [The discussion appropriately contextualises findings within the broader literature. However, the unexpected association between plant-based diets and increased disease activity requires further exploration]
Response 9: Thank you for this comment. We agree that the unexpected findings of this study require further exploration. Further longitudinal and clinical studies are needed to explore the relationship between dietary intake and disease activity and inflammation. Line 232 of the discussion highlights the need for further studies to expand upon the findings.
Comment 10: [Consider discussing the implications of low dietary fibre intake and its potential role in IBD management more extensively]
Response 10: Thank you for this suggestion. At Line 186 of the discussion we have included a sentence for further exploration of the need for participants in this cohort to increase dietary fibre intake in line with current dietary recommendations. We have included in the discussion the complexities of dietary fibre intake in people with IBD including recommendations to reduce fibre and difficulties with tolerating fibre (Line 179-180).
Comment 11: [The limitations section is well-developed but could be strengthened by discussing the cross-sectional designs inability to establish causality.]
Response 11: Thank you for this suggestion. See Line 261 of the conclusion for the discussion about the limitation of the study design.
Comment 12: [The conclusions align with the study objectives and findings. They effectively summarise the study's relevance but could include a stronger call for future longitudinal studies to establish causal links]
Response 12: Thank you for this suggestion. We have included a more detailed call for longitudinal studies that assess dietary intake and disease activity overtime at Line 274 of the conclusion.
Reviewer 3 Report
Comments and Suggestions for Authors
Well designed, conducted, analyzed and discussednutritional epidemiology study. The only methodologicalproblem this reviewer sees is the use of non-incidentcases. Many people with IBD may have had the diseasefor a long time and may have changed their diet becauseof the disease.
However, this was also highlighted bythe authors in the discussion, as a limitation of thestudy.
There are other small problems: 1) please, deleteline 49; 2) line 162: demographic and clinical data; 3) line 196: p<0.05; 4)line 210: ethical considerations?
Author Response
Comment 1: [Well designed, conducted, analyzed and discussednutritional epidemiology study. The only methodologicalproblem this reviewer sees is the use of non-incidentcases. Many people with IBD may have had the diseasefor a long time and may have changed their diet becauseof the disease.
However, this was also highlighted by the authors in the discussion, as a limitation of the study.
There are other small problems: 1) please, deleteline 49; 2) line 162: demographic and clinical data; 3) line 196: p<0.05; 4)line 210: ethical considerations?]
Response 1: Thank you kindly for reviewing this manuscript and for identifying the changes to be made. See below responses:
- Thank you for identifying this issue. The template text in line 49 has been removed.
- The heading at line 162 has been changed to "Demographic and clinical data".
- The symbol denoting the p-value has been corrected on line 196.
- The ethical considerations section has been added.